# *Cladosporium* Species Associated with Fruit Trees in Guizhou Province, China

**DOI:** 10.3390/jof9020250

**Published:** 2023-02-13

**Authors:** Yuanqiao Yang, Wenmei Luo, Wensong Zhang, Mohammed Amin Uddin Mridha, Subodini Nuwanthika Wijesinghe, Eric H. C. McKenzie, Yong Wang

**Affiliations:** 1Department of Plant Pathology, Agriculture College, Guizhou University, Guiyang 550025, China; 2Faculty of Graduate Studies, Daffodil International University, Birulia, Dhaka 1216, Bangladesh; 3Center of Excellence in Fungal Research, Mae Fah Luang University, Chiang Rai 57100, Thailand; 4Auckland Mail Centre, P.O. Box 92170 Auckland 1142, New Zealand

**Keywords:** 7 new taxa, asexual morph, Cladosporiaceae, hyphomycetes, taxonomy

## Abstract

During an investigation of fungal diversity on fruit trees in Guizhou Province, 23 *Cladosporium* strains were isolated from various locations in Guizhou Province. Culture characteristics, morphology and molecular phylogenetic analysis of three genetic markers, namely, the internal transcribed spacer regions (ITS) of the rDNA, partial fragments of actin (*act*), and the translation elongation factor 1-α (*tef*1-ɑ) loci were used to characterize these isolates. Seven new *Cladosporium* species and new host records for five other species were introduced, with detailed descriptions and illustrations. This study showed that there is a rich diversity of *Cladosporium* spp. in fruit trees in Guizhou Province.

## 1. Introduction

*Cladosporium* is accommodated in a large family, Cladosporiaceae, with its asexual morphs being dematiaceous hyphomycetes [1]. The conidia of *Cladosporium* usually form in branched chains and are so small that they can spread easily as one of the most common air-borne microorganisms [2,3,4]. This genus includes more than 878 epithets in the Index Fungorum database (https://www.indexfungorum.org/; accessed on 4 December 2022). The species are commonly discovered as endophytes, plant pathogens, human pathogens, and hyperparasites of other fungi [5,6,7,8,9]. A strain of *C. cladosporioides* may have potential as a biocontrol agent for reducing apple scab caused by *Venturia inaequalis* in leaves and fruit [10]. *Cladosporium* spp. can also produce compounds of medical interest or as potential biocontrol agents for other plant diseases [11,12].

In the past two decades, molecular phylogeny has widely entered the taxonomy of *Cladosporium* from single locus to multi-gene analysis. For example, Bensch et al. used three loci (internal transcribed spacer regions (ITS) of the rDNA, partial fragments of actin (*act*), and the translation elongation factor 1-α (*tef*1-ɑ) genes) to define species entities within the *C. cladosporioides* complex [13,14,15]. Subsequent researchers have often selected these genes to explain the phylogenetic relationship of *Cladosporium* spp. [6,7,13,14,15,16,17,18,19,20].

The present study uses three gene loci to establish a phylogenetic tree for our new strains and provides a morphological comparison and colony characteristics for 23 *Cladosporium* isolates obtained from fruit trees in Guizhou Province, China.

## 2. Materials and Methods

### 2.1. Sample Collection, Fungal Strain Isolation and Morphology

From 2021 to 2022, 94 samples were collected from orchards in six locations of Guizhou Province (Congjiang, Kaiyang, Longli, Luodian, Nayong and Wengan counties), including 10 host species (cherry, loquat, passionfruit, pitaya, plum, pomegranate, *Rosa roxburghii*, shaddock, tangerines and walnut). To obtain pure cultures, leaf surfaces were disinfected according to Zhang et al. [21]. Abundant conidia were observed on the surface of the leaf spots examined using a dissecting microscope. Single conidia were picked off the leaves with a sterilized needle and placed on a drip board containing sterilized water. After 12 h, the germination of conidium was observed, and they were then transferred to potato dextrose agar (PDA) and incubated at room temperature (28 °C) for 10 days. Morphological characteristics of the fungi were observed and photographed using a compound light microscope (Zeiss Scope 5) with an attached camera (AxioCam 208 color). All new taxa were registered in the Index Fungorum database (www.indexfungorum.org) (accessed on 13 October 2022). Dried holotype specimens were conserved in the Herbarium of the Department of Plant Pathology, Agricultural College, Guizhou University (HGUP), while cultures were conserved in the Departmental Culture Collection (GUCC).

### 2.2. DNA Extraction and PCR Amplification

Once the fungal colonies had grown to the edge of 90 mm diam. Petri dish, a sterile scalpel was used to transfer mycelium to a 1.5 mL centrifuge tube and the genomic DNA was extracted with PrepMan Ultra Reagent (Applied Biosystems, CA, USA) following the manufacturer’s protocol. PCR amplification was performed in a 25 μL reaction volume system. For *Cladosporium* isolates, the ITS region was first sequenced, and a BLAST search of GenBank was used to reveal the closest matching taxa. The *tef*1-ɑ, and *act* loci were also employed to support the species identification. Primers ITS4 and ITS5 were used to amplify the ITS [22], and EF-728F and EF-986R were used for *tef*1-ɑ [23]. The *act* region was amplified with primers ACT-512F and ACT-783R [23]. The PCR thermal cycle program for ITS, *tef*1-ɑ, and *act* amplification was: initial denaturation at 95 °C for 5 min, followed by 40 cycles of denaturation at 95 °C for 30 s, annealing at 54 °C for 30 s, elongation at 72 °C for 1 min and the final extension at 72 °C for 10 min. Purification and sequencing of the PCR amplicons were undertaken by Sangon Biotech (Chengdu, China). The qualified sequences were submitted to GenBank and their accession numbers are shown in Table 1. All strains used in this study are listed in Table 1.

### 2.3. Phylogenetic Analysis

Multiple sequence alignments were constructed and carried out using the MAFFT v.7.110 online program (http://mafft.cbrc.jp/alignment/server/) last accessed on 13 October 2022. The phylogeny website tool “ALTER” [24] was used to transfer the alignment file from “.nex” to “.phy” file for RAxML analysis. We chose “MAFFT” and “FASTA” in select format and then uploaded our alignment. Finally, we chose “PhyML” in the third step, which is to select output format and convert. Maximum likelihood (ML) analysis was performed at the CIPRES Science Gateway v. 3.3 (http://www.phylo.org/portal2/, last accessed on 13 October 2022 [25]) using RAxML v.8.2.8 as part of the “RAxML-HPC BlackBox” tool [26,27]. All free model parameters were estimated by RAxML with ML estimates of 25 per site rate categories. The final ML search was conducted using the GTRGAMMA+I model. Bayesian analysis (BI) was performed on the website of CIPRES Science Gateway v.3.3 platform (http://www.phylo.org/portal2/, last accessed on 13 October 2022 [25]) using MrBayes on XSEDE (3.2.7a) as a tool [26,27]. Maximum parsimony (MP) was computed in PAUP* 4.0b10 [28] with the default settings, using the heuristic search option with 1000 random sequence addition replicates and tree bisection-reconnection (TBR) as the branch swapping algorithm. Maxtrees was set at 10,000. The tree length (TL), consistency indices (CI), retention indices (RI), rescaled consistency indices (RC) and homoplasy index (HI) were calculated for each generated tree. Bayesian inference (BI), GTR+I+G was selected as the best model for all three loci (ITS, *tef*1-ɑ and *act*) as determined by MrModeltest v2 [29]. BI analysis was undertaken using MrBayes v. 3.2.6 [30]. Six Markov chain Monte Carlo were launched with random starting trees for 5,000,000 generations and sampled every 5000 generations. The first 25% of the resulting trees were discarded as burn-in. The final phylogenetic topology was viewed in FigTree [31] and edited in Adobe Illustrator CS5. Moreover, information on all downloaded strains used to construct the phylogenetic tree were listed in Table 2.

## 3. Results

### 3.1. Phylogenetic Analyses

The alignment of ITS-*tef*1-ɑ-*act* included 138 isolates, including two outgroup taxa, *Toxicocladosporium irritans* (CBS 185.58) and *T. protearum* (CBS 126499), which yielded 1198 characters (ITS: 1–266; *tef*1-ɑ: 267–817; *act*: 818–1198). For MP analysis, 661 were constants, 154 were variables and parsimony uninformative, and 383 were parsimony-informative characters (TL = 2988, CI = 0.32, RI = 0.68, RC = 0.22, HI = 0.68). In the ML analysis, the RAxML tree for the best score is given (Figure 1), with a final likelihood value of −16,305.936706. The matrix has 699 different alignment patterns, with 15.5% of unidentified characters or gaps. The estimated fundamental frequencies are: A = 0.227633, C = 0.292355, G = 0.249907, T = 0.230105; gamma distribution shape parameter alpha = 0.567453. The MP, ML and BI yielded similar topologies, and the ML one was selected and edited (Figure 1).

In the phylogenetic tree, all 96 known *Cladosporium* taxa and our 23 strains formed a strong clade (100% ML/100% MP/1 PP). Our fungal strains were scattered in six phylogenetic Clades (Figure 1). Over half of our strains (12) belonged to clade 1 (0.96 PP), which included *C. tenuissimum* and five related taxa. Among them, all branches had high statistical support, except for GUCC 21265.2, perhaps due to the root of this clade. Six strains (GUCC 21206.1, GUCC 21266.1, GUCC 21208.3, GUCC 21208.5, GUCC 21289.5 and GUCC 21265.2) should represent three independent units. Clade 2 included four of our strains (GUCC 21244.1, GUCC 21259.1, GUCC 21227.4 and GUCC 21220.1). Strains GUCC 21244.1 and GUCC 21259.1 formed a branch (98% ML/98% MP/1 PP). GUCC 21227.4 and the ex-type culture of *C. guizhouense* (GUCC 401.8) clustered together, with high support values (91% ML/98% MP/1 PP). GUCC 21220.1 formed a close relationship with *C. puris* (COAD 294) (98% ML/99% MP/1 PP). Only one of our strains (GUCC 21262.1) in clade 3 had no phylogenetic distance from the ex-type strain of *C. eucommiae*. Clade 4 included two taxa (*C. cucumerinum* and *C. subuliformae*) and three of our strains (GUCC 21212.1, GUCC 21208.1 and GUCC 21208.2), which formed a branch (99% ML/98% MP/1 PP) with an ex-type culture of *C. subuliformae*. For clade 5, strain GUCC 21260.3 had a close relationship to *C. xylophilum* (CBS 125997T and CBS 113749) (99% ML/99% MP/1 PP), but with some phylogenetic distance. The two remaining strains (GUCC 21267.1 and GUCC 21271.5) were accommodated in clade 6, with GUCC 21267.1 being *C. xanthochromaticum* and GUCC 21271.5 forming a branch with an ex-type strain of *C. perangustum* (CBS 125996) (94% ML/100% MP/1 PP). The DNA base differences on different gene loci between our *Cladosporium* strains and their relatives are summarized in Table 3.

### 3.2. Morphology and Culture Characteristics

***Cladosporium pruni-salicina*** Y.Q. Yang & Yong Wang bis, sp. nov. (Figure 2)

Index Fungorum number: IF900107

**Etymology**: *pruni-salicina*, in reference to the host plant (*Prunus salicina*).

**Sexual morph**: Not observed. **Asexual morph**: hyphomycetous. ***Mycelium*** superficial and immersed, abundant, composed of septate, branched hyphae, overgrowing entire culture dish, hyaline, smooth or almost so, slightly curved, 1–3 µm wide. ***Conidiophores*** 9.0–124.0 × 2.5–5.0 µm (x = 44.1 × 3.8 µm; *n* = 20), solitary or in small loose groups, erect to slightly flexuous, slightly thickened toward the apex, pale olivaceous gray, nodulose and thick-walled. ***Conidia*** 3.0–6.5 × 2.0–4.0 µm (x = 4.1 × 2.8 µm; *n* = 30), solitary or in short unbranched chains, mostly light gray to yellow, aseptate, cylindrical-oblong, nodulose and thin-walled, variable in size and shape, subglobose, ellipsoid-ovoid, obovoid, fusiform, subcylindrical. ***Secondary ramoconidia*** 4.5–9.0 × 2.5–4.5 µm (x= 6.1 × 3.3 µm; *n* = 30), pale to medium olivaceous brown, subcylindrical to cylindrical, aseptate, thin-walled.

**Culture characteristics:*****Colonies*** on PDA reaching 48–61 mm diam. after 2 weeks at 25 ℃, light yellowish-brown, margin regular and pale yellow, reverse dull-yellow, flat, velvety, diffuse. ***Colonies*** on MEA reaching 45–60 mm diam. after 2 weeks at 25 °C, light brown and dull-tan margin, abundant, flat or low convex, feathery, submerged margin, radially furrowed. ***Colonies*** on SNA reaching 38–45 mm diam. after 2 weeks at 25 °C, gray-olivaceous to olivaceous, olivaceous-gray reverse, flat, velvety, sparse, margin regular. Without prominent exudates, sporulation profuse on all media.

**Material examined:** China, Guizhou Province, Kaiyang County, on eaves of *Prunus salicina* Lindl., June 2021, Y.Q. Yang (HGUP 21206, holotype); ex-type living culture GUCC 21206.1. Ibid. (HGUP 21266), living culture GUCC 21266.1.

**Notes:** Two strains (GUCC 21206.1 and GUCC 21266.1) representing *C. pruni-salicina* formed an independent branch in clade 1 (Figure 1) but also displayed a relatively close distance to ex-type cultures of *C. colocasiae* (CBS 386.64) and *C. oxysporum* (CBS 125991). *Cladosporium pruni-salicina* can be distinguished from *C. colocasiae* on the DNA base differences of ITS, *tef*1-ɑ and *act* loci (1/535 in ITS, 30/318 in *tef*1-ɑ and 12/232 in *act*). The base differences with *C. oxysporum* are (1/540 in ITS, 21/295 in *tef*1-ɑ and 16/226 in *act*) (Table 3). Conidia (3.0–6.5 × 2.0–4.0 µm) and secondary ramoconidia (4.5–9.0 × 2.5–4.5 µm) of *C. pruni-salicina* are obviously narrower and shorter than those of *C. colocasiae* (9–16 × 5–7(–8) µm). The conidia are longer and broader than those of *C. oxysporum* (3.0–5.0 × 2.0–3.0), while the secondary ramoconidia are shorter than those of *C. oxysporum* (7.0–21.0 × 3.0–4.0). The conidiophores of *C. pruni-salicina* are usually shorter than those of *C. colocasiae* and *C. oxysporum* (9.0–124.0 µm vs. 110–180 µm vs. 30–115 µm). Colonies of *C. pruni-salicina* on PDA are light yellowish-brown with pale yellow margins, while *C. colocasiae* colonies are gray-olivaceous to olivaceous or dull green [15]. Thus, *Cladosporium pruni-salicina* is introduced as a distinct novel taxon.

***Cladosporium congjiangedsis*** Y.Q. Yang & Yong Wang bis, *sp.*
*nov.* (Figure 3)

Index Fungorum number: IF900108

**Etymology:***congjiangedsis*, in reference to the location from which the fungus was isolated.

**Sexual morph:** Not observed. **Asexual morph:** hyphomycetous. ***Mycelium*** superficial and immersed, abundant, composed of septate, branched hyphae, overgrowing entire culture dishes, pale or medium olivaceous brown, nodulose, smooth or almost so, slightly curved, often with swellings and constrictions, 1.5–4 µm wide. ***Conidiophores*** 15.5–103.0 × 2.5–5.0 µm (x = 56.5 × 3.4 µm; *n* = 20), solitary or in small loose groups, straight or somewhat flexuous, slightly thickened toward the apex, cylindrical-oblong, pale olivaceous brown, nodulose and thick-walled. ***Conidia*** 2.5–5.5 × 1.5–4.0 µm (x = 4.1 × 2.9 µm; *n* = 30), solitary or in short unbranched chains, subglobose to obovoid, obovoid, limoniform or ellipsoid, aseptate, protuberant hila, pale or medium olivaceous brown. ***Secondary ramoconidia*** 4.5–9.5 × 2.5–5.5 µm (x = 6.6 × 4.6 µm; *n* = 30), medium to dark brown, oblong, oblong-ellipsoid, subcylindrical to cylindrical, narrowed base, often constricted at septum, thin-walled.

**Culture characteristics:*****Colonies*** on PDA reaching 45–58 mm diam. after 2 weeks at 25 °C, pale yellowish-brown, margin dark brown, flat, black reverse, velvety, diffuse. ***Colonies*** on MEA reaching 46–58 mm diam. after 2 weeks at 25 °C, dull-brown toward the center, velvety or powdery, abundant, growth flat to low convex, reverse black, radially furrowed, feathery, submerged margin. ***Colonies*** on SNA reaching 45–59 mm diam. after 2 weeks at 25 °C, light brown to dark brown, flat, velvety, sparse, margin regular, aerial mycelium loose diffuse, pale yellow. Without prominent exudates, sporulation profuse on all media.

**Material examined:** China, Guizhou Province, Congjiang County, on leaves of *Passiflora edulis* Sims, August 2021, Y.Q. Yang (HGUP 21208, holotype); ex-type living culture GUCC 21208.3 = GUCC 21208.5; China, Guizhou Province, Luodian County, on leaves of *Citrus maxima* (Burm.) Merr. (HGUP 21289), September 2021, Y.Q. Yang, living culture GUCC 21289.5.

**Notes:** Phylogenetically, three strains (GUCC 21208.3, GUCC 21208.5 and GUCC 21289.5) formed a branch with support values of (ML/MP/BI = 92/91/1) and clustered with *C. tenuissimum*, *C. colocasiae*, *C. oxysporum* and *C. pruni-salicina* with moderate support (Figure 1). The basal differences with ex-type cultures of *C. tenuissimum* (CBS 125995), *C. colocasiae* (CBS 386.64), *C. oxysporum* (CBS 125991) and *C. pruni-salicina* (GUCC 21206.1) were as follows (0/538; 0/536; 1/538; 1/538 for ITS, 17/235; 22/318; 16/235; 28/235 for *tef*1-ɑ and 8/232; 14/232; 18/232; 4/232; for *act*) (Table 3). Morphologically, the conidia of *C. congjiangedsis* are globose or obovoid rather than long and cylindrical, similar to *C. colocasiae*. The conidia of *C. congjiangedsis* (2.5–5.5 × 1.5–4.0 µm) are similar in size to those of *C. oxysporum* (3–5 × 2–3 µm) and *C. tenuissimum* (2.5–5 × 2–3 µm). Both *C. oxysporum* and *C. tenuissimum* show smoke-gray to pale olivaceous-gray colonies, while those of *C. congjiangedsis* are pale yellowish-brown [15]. The conidia of *C. congjiangedsis* are slightly smaller compared to *C. pruni-salicina* (3–6.5 × 2–4) and the latter has an obvious radial furrow in the colony morphology, but *C. congjiangedsis* does not. Therefore, *Cldosporium congjiangedsis* is presented as a novel taxon.

*****Cladosporium* kaiyangensis**** Y.Q. Yang & Yong Wang bis, *sp. nov.* (Figure 4)

Index Fungorum number: IF900109

**Etymology**: *kaiyangensis*, in reference to the location where the fungus was isolated.

**Sexual morph:** Not observed. **Asexual morph:** hyphomycetous. ***Mycelium*** superficial and immersed, composed of unbranched or loosely branched hyphae, septate, pale or medium olivaceous brown, smooth or almost so, minutely verruculose or irregularly rough-walled, walls slightly thickened, 2–3.5 µm wide. ***Conidiophores*** 23.5–123.0 × 3.0–7.0 µm (x = 58.5 × 3.5 µm; *n* = 20), macro- and micronematous, formed solitary or in groups of three laterally or terminally from hyphae, straight or somewhat flexuous, neither geniculate nor nodulose, cylindrical-oblong. ***Conidia*** 3.5–6.5 × 2.5–3.5 µm (x = 5.0 × 2.9 µm; *n* = 30), subglobose, obovoid or ellipsoid, occasionally globose, limoniform or short ellipsoid, pale or medium olivaceous brown. ***Secondary ramoconidia*** 4.5–14.0 × 2.5–4.0 µm (x = 6.9 × 3.2 µm; *n* = 30), medium to dark brown, oblong, oblong-ellipsoid, ellipsoid to cylindrical, smooth- and thin-walled, with a protuberant, somewhat darkened, narrowed base, aseptate.

**Culture characteristics:*****Colonies*** on PDA reaching 48–62 mm diam. after 2 weeks at 25 °C, olivaceous brown or pale olivaceous brown, dull-yellow toward the margins, reverse deep black-yellow, dull yellow toward the margins, fluffy-felty, margin broad, feathery, somewhat undulate, aerial mycelium abundant, loose to dense, low to high, sporulating. ***Colonies*** on MEA reaching 47–55 mm diam. after 2 weeks at 25 °C, olivaceous brown at margins where sporulation is profuse, reverse dark brown, fluffy-felt, margin pale yellow, feathery, aerial mycelium abundant, loose to high, colony center folded and wrinkled, radially furrowed, without prominent exudates. ***Colonies*** on SNA reaching 48–61 mm diam. after 2 weeks at 25 °C, light brown to dark brown, flat, velvety, sparse, margin regular, aerial mycelium loose diffuse, pale yellow and uniform and regular color. Without prominent exudates, sporulation profuse on all media.

**Material examined:** China, Guizhou Province, Kaiyang County, on decaying fruit of *Eriobotrya japonica* (Thunb.) Lindl., April 2021, Y.Q. Yang (HGUP 21265, holotype); ex-type living culture GUCC 21265.2.

**Notes:** Phylogenetically, *Cladosporium kaiyangensis* was placed in the root of clade 1 but only with the BP value (0.96) (Figure 1). *Cladosporium kaiyangensis* is morphologically comparable with *C. colocasiae*, but it has generally shorter conidiophores (23.5–123.0 × 3.0–7.0 vs. 110–180 × 4–6 µm), slightly shorter and narrower secondary ramoconidia and conidia, and somewhat wider conidiogenous loci and hila. The conidia of *C. kaiyangensis* are narrower than those of *C. tenuissimum*, *C. colocasiae*, *C. oxysporum*, *C. pruni-salicina* and *C. congjiangedsis*, which are all located close to each other in the phylogenetic tree. *Cladosporium* colocasiae, *C. tenuissimum* and *C. oxysporum* have gray-olivaceous to olivaceous or dull green colonies on PDA [15], while those of *C. kaiyangensis* are olivaceous brown or pale olivaceous brown with dull-yellow toward the margins. Additionally, the margin of *C. kaiyangensis*’s colony did not have a light white ring or dark brown ring similar to *C. pruni-salicina* and *C. congjiangedsis*, nor was there an obvious radial furrow. *Cladosporium kaiyangensis* can be distinguished from ex-type cultures of *C. tenuissimum* (CBS 125995), *C. colocasiae* (CBS 386.64), *C. oxysporum* (CBS 125991), *C. pruni-salicina* (GUCC 21206.1) and *C. congjiangedsis* (GUCC 21208.3) based on ITS, *tef*1-ɑ and *act* loci (4/540, 7/540, 4/540, 5/540, 4/540 in ITS, 25/249, 28/249, 21/249, 35/249, 19/249 in *tef*1-ɑ and 3/232, 16/232, 18/232, 4/232, 4/232 in *act*) (Table 3). Thus, we propose *Cladosporium kaiyangensis* as a phylogenetically distinct species.

***Cladosporium ribus*** Y.Q. Yang & Yong Wang bis, *sp. nov.* (Figure 5)

Index Fungorum number: IF900110

**Etymology:***ribus*, in reference to the plant genus (*Ribes burejense*), from which the fungus was isolated.

**Sexual morph:** Not observed. **Asexual morph:** hyphomycetous. ***Mycelium*** superficial and immersed, with abundant, filiform or narrowly cylindrical, branched, septate hyphae, neither swollen nor constricted, subhyaline or pale olivaceous, almost smooth, asperulate or loosely verruculose, especially those hyphae forming conidiophores with surface ornamentation, 1.5–4 µm wide. ***Conidiophores*** 8.0–103.5 × 2.0–4.0 µm (x = 46.1 × 3.0 µm; *n* = 20), macro- and micronematous, arising terminally or laterally from plagiotropous or ascending hyphae, macronematous conidiophores narrowly cylindrical-oblong, often distinctly geniculate, sometimes growth proceeding at an angle of 45−90°, subnodulose, sometimes forming lateral shoulders at or towards the apex, mostly unbranched. ***Conidia*** 2.5–4.5 × 2.0–3.0 µm (x = 3.4 × 2.3 µm; *n* = 30), ellipsoid, aseptate, very pale olivaceous, unthickened but somewhat refractive, numerous, apex rounded, ellipsoid or subcylindrical. ***Secondary ramoconidia*** 4.0–8.5 × 2.0–3.5 µm (x = 5.3 × 2.6 µm; *n* = 30), ellipsoid, subcylindrical or cylindrical, pale olivaceous or pale to medium olivaceous brown, smooth, occasionally slightly rough-walled, walls unthickened or almost so, hila conspicuous, subdenticulate or denticulate.

**Culture characteristics:*****Colonies*** on PDA reaching 50–65 mm diam. after 2 weeks at 25 °C, pale olivaceous brown, yellow toward the margins, velvety, margin broad, white, regular, glabrous to feathery, aerial mycelium absent or sparse, growth regular, low convex. ***Colonies*** on MEA reaching 46–58 mm diam. after 2 weeks at 25 °C, pale olivaceous yellow, reverse dark brown inside extending to yellowish edges, white, floccose or fluffy-felty, margin regular, feathery, aerial mycelium whitish, abundant, growth effuse, flat or low convex, undulate, submerged margin. ***Colonies*** on SNA reaching 45–60 mm diam. after 2 weeks at 25 °C, pale yellow-brown, light-brown at margins, reverse pale yellow or pale olivaceous brown, floccose or felty, margins regular, glabrous, aerial mycelium covering large parts. Without prominent exudates, sporulation profuse on all media.

**Material examined:** China, Guizhou Province, Longli County, on leaves of *Ribes burejense* Fr. Schmidt, June 2021, Y.Q. Yang (HGUP 21244, holotype); ex-type living culture GUCC 21244.1; China, Guizhou Province, Nayong County, on leaves of *Prunus pseudocerasus* (Lindl.) G. Don, March 2021, Y.Q. Yang (HGUP 21259), living culture GUCC 21259.1.

**Notes:** Two strains of *Cladosporium ribus* (GUCC 2124.1 and GUCC 21259.1) displayed a close relationship with *C. guizhouense* (GUCC 401.8, ex-type culture and GUCC 21227.4) in clade 2. *Cladosporium ribus* produces similar-sized conidia to *C. guizhouense* (2.5–4.5 × 2.0–3.0 µm vs. 3–7.5 × 2.5–4 µm) but its secondary ramoconidia are generally smaller (4.0–8.5 × 2.0–3.5 µm vs. 6.5–23 × 3–5.5 µm). *Cladosporium ribus* can be distinguished from *C. guizhouense* based on ITS, *tef*1-ɑ and *act* loci (0/537 in ITS, 20/245 in *tef*1-ɑ and 4/232 in *act*) (Table 3) [20]. Thus, we identified *Cladosporium ribus* as a new species.

*****Cladosporium* wenganensis**** Y.Q. Yang & Yong Wang bis, *sp. nov.* (Figure 6)

Index Fungorum number: IF900111

**Etymology:***wenganensis*, in reference to the location from where the fungus was isolated.

**Sexual morph:** Not observed. **Asexual morph:** hyphomycetous. ***Mycelium*** superficial and immersed, hyphae branched, septate, subhyaline, pale olivaceous or pale olivaceous brown, obviously nodulose or slightly rough-walled, thin-walled, sometimes forming ropes, occasionally swollen at the base of conidiophores, 3.5–5.0 µm wide. ***Conidiophores*** 11.5–151.5 × 3.5–5.5 µm (x = 45.2 × 4.6 µm; *n* = 20), macronematous, solitary, cylindrical, cylindrical-oblong or irregular in outline due to swellings and constrictions, subnodulose, straight or often somewhat flexuous, formed laterally or terminally from hyphae, 0–4-septate, not constricted at septa, pale to medium olivaceous brown, smooth or almost so, walls slightly thickened. ***Conidia*** 3.0–5.5 × 2.0–4.0 µm (x = 4.3 × 3.3 µm; *n* = 30), solitary or formed in short chains, ellipsoid, broadly ovoid or subcylindrical, limoniform, aseptate, very pale olivaceous, walls unthickened but somewhat refractive, numerous, apex rounded. ***Secondary ramoconidia*** 4.0–8.0 × 3.0–5.0 µm (x = 5.8 × 4.0 µm; *n* = 30), ellipsoid, subcylindrical or cylindrical, pale olivaceous or pale to medium olivaceous brown, obviously nodulose, occasionally slightly rough-walled, walls unthickened, hila conspicuous, subdenticulate or denticulate.

**Culture characteristics:*****Colonies*** on PDA reaching 50–65 mm diam. after 2 weeks at 25 °C, smoke-gray and olivaceous due to abundant and dense aerial mycelium, olivaceous gray toward the margins, reverse dull olive-brown, with a whitish narrow final edge, fluffy, margins narrow, white, somewhat feathery, regular or slightly undulate, growth flat, sporulation loose. ***Colonies*** on MEA reaching 40–52 mm diam. after 2 weeks at 25 °C, smoke-gray to light olive-gray due to abundant aerial mycelium, reverse dull olive-brown, velvety or fluffy, with a dark olivaceous brown narrow final edge. ***Colonies*** on SNA reaching 45–55 mm diam. after 2 weeks at 25 °C, gray-olivaceous to olivaceous, olivaceous-gray reverse, flat, velvety, sparse mycelium, margin regularly. Without prominent exudates, sporulation profuse on all media.

**Material examined:** China, Guizhou Province, Wengan County, on leaves of *Prunus persica* L., June 2021, Y.Q. Yang (HGUP 21220, holotype); ex-type living culture GUCC 21220.1.

**Notes:** Phylogenetically, *Cladosporium wenganensis* (GUCC 21220.1) was sister to *C. puris* (COAD 2494) with high statistical support (ML/MP/BI = 98/99/1) (Figure 1). The comparison of DNA base composition (Table 3) indicated that between C. wenganensis and *C. puris*, there was only 1 base difference in the ITS region, but 29 base differences in the *tef*1-ɑ region and 7 base differences in the *act* region. *Cladosporium wenganensis* had wider and shorter conidiophores (11.5–151.5 × 3.5–5.5 µm vs. 44–225 × 2–3 µm) than *C. puris*, as well as slightly wider secondary conidia (4.0–8.0 × 3.0–5.0 µm vs. 5–12.5 × 2–3.5 µm) and slightly larger conidia (3.0–5.5 × 2.0–4.0 µm vs. 2.5–4.5 × 2–3 µm) [32]. *Cladosporium wenganensis* was introduced as a new species.

***Cladosporium nayongensis*** Y.Q. Yang & Yong Wang bis, *sp. nov.* (Figure 7)

Index Fungorum number: IF900112

**Etymology:***nayongensis*, in reference to the location where the fungus was isolated.

**Sexual morph:** Not observed. **Asexual morph:** hyphomycetous. ***Mycelium*** superficial and immersed, hyphae unbranched or sparingly branched, septate, sometimes constricted at septa, especially in wider ones, subhyaline to pale olivaceous or pale olivaceous brown, obviously nodulose or almost so, 2.0–4.0 µm wide, walls sometimes slightly thickened. ***Conidiophores*** 16.5–110.0 × 2.0–4.5 µm (x = 61.2 × 2.9 µm; *n* = 20), macronematous, solitary, cylindrical, cylindrical-oblong or irregular in outline due to swellings and constrictions, subnodulose, straight or often somewhat flexuous, formed laterally or terminally from hyphae, unbranched or branched once or twice, occasionally three times, branches often only as short denticle-like lateral outgrowths just below a septum, walls slightly thickened. ***Conidia*** 3.5–5.5 × 2.0–3.5 µm (x = 4.4 × 2.7 µm; *n* = 30), numerous, solitary or formed in short chains, obovoid, ovoid to limoniform or ellipsoid, sometimes subglobose, aseptate, pale olivaceous brown, unthickened but somewhat refractive, numerous, apex rounded. ***Secondary ramoconidia*** 4.0–7.5 × 2.0–3.5 µm (x = 5.7 × 2.8 µm; *n* = 30), ovoid to subcylindrical or cylindrical-oblong, pale olivaceous or pale to medium olivaceous brown, obviously nodulose, sometimes slightly rough-walled, walls unthickened, and darkened-refractive.

**Culture characteristics:*****Colonies*** on PDA reaching 55–68 mm diam. after 2 weeks at 25 °C, dull yellow brown due to abundant and dense aerial mycelium, dull brown to pale yellow brown, with white margins, reverse dull olive-brown, with a pale yellow narrow final edge, fluffy, margins narrow, somewhat feathery, regular or slightly undulate, growth flat, somewhat radially furrowed, sporulation loose. ***Colonies*** on MEA reaching 52–65 mm diam. after 2 weeks at 25 °C, pale yellow brown due to abundant aerial mycelium, reverse dull olive-brown, velvety or fluffy, with a pale white-yellow narrow final edge, glabrous to somewhat feathery, aerial mycelium brown, floccose, abundant, dense, somewhat radially furrowed. ***Colonies*** on SNA reaching 65–78 mm diam. after 2 weeks at 25 °C, pale yellow, flat, velvety, margin regular, growth effuse to low convex, reverse light yellow. Without prominent exudates, sporulation profuse on all media.

**Material examined:** China, Guizhou Province, Nayong County, Leaves of *Prunus pseudocerasus* (Lindl.) G. Don, March 2021, Y. Q. Yang (HGUP 21260, holotype); ex-type living culture GUCC 21260.3;

**Notes:** The placement of *Cladosporium nayongensis* was close to *C. xylophilum* (CBS 125997 ex-type culture and CBS 113749), with high statistical support (ML/MP/BI = 99/99/1) in Figure 1. The comparison of DNA base composition (Table 3) indicated that between GUCC 21260.3 and CBS 125997, there were identical sequences in the ITS region, but 2 base differences in the *tef*1-ɑ region and 10 base differences in the *act* region. *Cladosporium xylophilum* sometimes produces numerous small, prominent exudates, but *C. nayongensis* does not. *Cladosporium nayongensis* has slightly smaller secondary conidia (4.0–7.5 × 2.0–3.5 vs. 7–23 × 2.5–4 µm) and wider conidia (3.5–5.5 × 2.0–3.5 µm vs. 2–5 × 2–2.5) than *C. xylophilum* [15]. Thus, *Cladosporium nayongensis* was introduced as a novel species.

***Cladosporium punicae*** Y.Q. Yang & Yong Wang bis, *sp. nov.* (Figure 8)

Index Fungorum number: IF900113

**Etymology:***punicae*, in reference to the host plant (*Punica granatum*), from which the fungus was isolated.

**Sexual morph:** Not observed. **Asexual morph:** hyphomycetous. ***Mycelium*** superficial and immersed, hyphae unbranched or very sparingly branched, subhyaline to pale or medium olivaceous-brown, smooth to minutely verruculose or irregularly verrucose, walls unthickened, sometimes forming ropes, 2.0–4.5 µm wide. ***Conidiophores*** 32.0–135.5 × 2.0–4.5 µm (x = 64.9 × 2.3 µm; *n* = 20), macro- and micronematous, solitary, arising terminally or laterally from plagiotropous or ascending and erect hyphae, erect, straight to slightly flexuous, cylindrical-oblong, sometimes slightly geniculate toward the apex, nodulose, pale to medium olivaceous-brown, paler toward the apex and sometimes attenuated, smooth to asperulate or minutely verruculose, walls slightly thickened. ***Conidia*** 2.5–5.5 × 2.0–3.5 µm (x = 3.3 × 2.5 µm; *n* = 30), numerous, obovoid, ovoid, fusiform to ellipsoid, fusiform, subcylindrical, subhyaline to pale olivaceous-brown, smooth to minutely verruculose or irregularly rough-walled. ***Secondary ramoconidia*** 3.0–7.0 × 2.0–3.5 µm (x = 4.6 × 2.7 µm; *n* = 30), pale olivaceous brown, smooth to minutely verruculose or irregularly verrucose, thickened and darkened-refractive, slightly attenuated toward the apex and base, hila subdenticulate or denticulate, protuberant.

**Culture characteristics:*****Colonies*** on PDA reaching 58–70 mm diam. after 2 weeks at 25 °C, pale yellow brown due to abundant and dense aerial mycelium, reverse dull olive-brown, with a pale yellow narrow final edge, fluffy, margins narrow, feathery, aerial mycelium loose, diffuse, growth flat, radially furrowed. ***Colonies*** on MEA reaching 60–75 mm diam. after 2 weeks at 25 °C, pale yellow brown due to abundant aerial mycelium, reverse dull olive-brown, sometimes yellowish white at margins, margins narrow, glabrous or feathery, radially furrowed, folded and wrinkled in colony center, aerial mycelium sparse, diffuse. ***Colonies*** on SNA reaching 46–58 mm diam. after 2 weeks at 25 °C, pale yellow-brown, flat, powdery to felty-floccose, velvety, margin regularly, growth effuse to low convex, reverse light yellow. Without prominent exudates, sporulation profuse on all media.

**Material examined:** China, Guizhou Province, Wengan County, on leaves of *Punica granatum* L., June 2021, Y.Q. Yang (HGUP 21271, holotype); ex-type living culture GUCC 21271.5.

**Notes:** The phylogenetic position of *Cladosporium punicae* (GUCC 21271.5) was sister to the ex-type culture of *C. perangustum* (CBS 125996) with high statistical support (ML/MP/BI = 94/100/1) (Figure 1). The comparison of DNA base composition indicated that between GUCC 21271.5 and CBS 125996, there were only 2 base differences in the ITS region, but 31 base differences in the *tef*1-ɑ region and 7 base differences in the *act* region (Table 3). The colony of *C. punicae* is pale yellow-brown on PDA and dull olive-brown on MEA, but *C. perangustum* is olivaceous-gray or iron-gray on PDA and pale olivaceous-gray to glaucous-gray or dull gray-olivaceous on MEA. Additionally, *C. punicae* produces slightly wider conidia and secondary conidia than *C. perangustum* [15]. Thus, *Cladosporium punicae* was proposed as a new species.

*Cladosporium subuliforme* Bensch, Crous & U. Braun, Studies in Mycology 67: 77 (2010) (Figure 9)

MycoBank No: 517090

**Sexual morph:** Not observed. **Asexual morph:** hyphomycetous. ***Mycelium*** superficial and immersed, subhyaline to pale olivaceous brown, smooth to minutely verruculose or verruculose, often somewhat swollen at the base of conidiophores, sometimes forming ropes, 2.5–5.5 µm wide. ***Conidiophores*** 10.5–114.0 × 2.5–5.5 µm (x = 39.9 × 4.0 µm; *n* = 20), solitary or in pairs, unbranched or branched, erect, straight to mostly flexuous, filiform to narrowly cylindrical-oblong, not nodulose or geniculate. ***Conidia*** 3.0–7.0 × 2.0–4.0 µm (x = 4.5 × 3.0 µm; *n* = 30), numerous, in branched chains, obovoid, subglobose, ovoid to limoniform or ellipsoid, often nodulose. ***Secondary ramoconidia*** 4.5–10.5 × 2.0–4.5 µm (x = 6.9 × 3.2 µm; *n* = 30), pale brown or pale olivaceous-brown, ellipsoid to subcylindrical, sometimes cylindrical-oblong, septate, median or somewhat in the lower half, usually somewhat attenuated toward the base.

**Culture characteristics:*****Colonies*** on PDA reaching 65–80 mm diam. after 2 weeks at 25 °C, mainly dull yellow brown, with yellowish white margins, margins broad, reverse dull olive-brown, fluffy, radially furrowed, feathery, aerial mycelium loose, diffuse, growth flat. ***Colonies*** on MEA reaching 64–78 mm diam. after 2 weeks at 25 °C, olivaceous buff due to abundant aerial mycelium, yellowish white at margins, reverse dull olivaceous brown, margins broad, glabrous, floccose to fluffy, aerial mycelium abundant, fluffy, mainly in colony center, diffuse. ***Colonies*** on SNA reaching 62–75 mm diam. after 2 weeks at 25 °C, pale yellow-brown, flat, powdery to felty-floccose, velvety, margin regular, growth effuse to low convex, reverse light yellow. Without prominent exudates, sporulation profuse on all media.

**Material examined:** China, Guizhou Province, Congjiang County, on leaves of *Passiflora edulis* Sims, August 2021, Y.Q. Yang, living culture GUCC 21208.1 = GUCC 21208.2; China, Guizhou Province, Dejiang County, on leaves of *Juglans regia* L., August 2021, Y.Q. Yang, living culture GUCC 21212.1 (new substrate record).

**Notes:** Our three strains (GUCC 21208.1, GUCC 21208.2, GUCC 21212.1) clustered with the ex-type culture of *C. subuliforme* (CBS 126500) with high statistical support (ML/MP/BI = 99/98/1). The comparison of DNA bases composition (Table 3) indicated that GUCC 21208.1, GUCC 21208.2, GUCC 21212.1 and CBS 126500 had identical sequences in the *act* region and ITS region, and only 1 base difference in the *tef*1-ɑ region. Morphologically, these three strains shared almost the same colony morphology, apical conidia and secondary conidia as the original description [15]. We concluded that the strain belonged to *C. subuliforme* as a new Chinese record on both hosts *Passiflora edulis* Sims and *Juglans regia* L.

*Cladosporium xanthochromaticum* Sand.-Den., Gené & Cano, Persoonia 36: 295 (2016) (Figure 10)

MycoBank No: 817340

**Sexual morph:** Not observed. **Asexual morph:** hyphomycetous. ***Mycelium*** superficial and immersed, septate, subhyaline, branched, pale olivaceous or pale olivaceous brown, thin to dense, hyphae straight to slightly sinuous, smooth or slightly rough-walled, sometimes appearing thin-walled, sometimes forming ropes, 1.5–4.5 µm wide. ***Conidiophores*** 19.0–130.5 × 2.5–4.5 µm (x = 52.0 × 3.1 µm; *n* = 20), septate, erect, solitary, sometimes distinctly constricted at septa, filiform or narrowly cylindrical-oblong, non-nodulose, occasionally once geniculate, walls slightly thickened. ***Conidia*** 2.5–6.5 × 2.0–3.5 µm (x = 4.6 × 2.6 µm; *n* = 30), numerous, aseptate, mostly light olive, variable in size and shape, obovoid, limoniform or short ellipsoid, often nodulose. ***Secondary ramoconidia*** 4.0–9.0 × 2.0–4.0 µm (x = 5.7 × 2.6 µm; *n* = 30), pale olivaceous brown, smooth- and thin-walled, somewhat darkened, cylindrical-oblong, walls unthickened or almost so, verruculose, attenuated toward the apex and base.

**Culture characteristics:*****Colonies*** on PDA reaching 58−70 mm diam. after 2 weeks at 25 °C, gray olivaceous or olivaceous, with yellowish-white margins, margin narrowed, reverse dull olive-brown, with yellow margins, margin broad, regular, glabrous to feathery, fluffy, aerial mycelium loose, diffuse, growth flat. ***Colonies*** on MEA reaching 40−55 mm diam. after 2 weeks at 25 °C, olivaceous, brown olivaceous and pale whitish-yellow toward the margins, reverse dull olive-brown, margin broad, radially furrowed, with raised, crater-shaped colony center, with white, undulate, submerged margin. ***Colonies*** on SNA reaching 45−62 mm diam. after 2 weeks at 25 °C, dull olive-brown, flat, powdery to felty-floccose, velvety, margin regular, growth effuse to low convex, reverse light yellowish. Without prominent exudates, sporulation profuse on all media.

**Material examined:** China, Guizhou Province, Luodian County, on leaves of *Hylocereus undatus* ‘Foo-Lon’, September 2021, Y.Q. Yang, living culture GUCC 21267.1 (new substrate record).

**Notes:** Our strain (GUCC 21267.1) formed a branch of the ex-type strain of *C. xanthochromaticum* (CBS 140691) with high statistical support (ML/MP/BI = 100/100/1) in Figure 1. The comparison of DNA base composition (Table 3) indicated that GUCC 21267.1 and CBS 140691 had identical sequences in the *act* region and only three base differences in the *tef*1-ɑ region and three in the ITS region. In morphology, GUCC 21267.1 shares almost the same colony morphology, apical conidia and secondary conidia as the description of *C. xanthochromaticum* [17]. Thus, GUCC 21267.1 is a new Chinese record of *Hylocereus undatus* ‘Foo-Lon’.

## 4. Discussion

Twenty-three *Cladosporium* isolates were obtained from ten plant hosts in six counties in Guizhou Province. Phylogenetic analyses indicated that they belonged to six clades (Clade 1 to Clade 6), which did not suggest powerful host speciation. The most prominent morphological feature typical of *Cladosporium* spp. is a thick refractive to darkened cladosporioid or coronate scar, defined as a raised periclinal rim with a central convex dome. The conidia are frequently 2–3-septate, regularly verrucose, short conidial chains and pronounced prolongations of the conidiophores [6,7,14,15,16]. However, in the present study, the morphological features of *Cladosporium* spp. were insufficient to support taxonomic conclusions. Thus, the taxonomy of this group needs phylogenetic analyses.

Seven novel species were introduced along with five other taxa that were newly recorded on various hosts, mainly based on phylogenetic analyses of three gene loci. Interestingly, all of our taxa belonged to the *C. cladosporioides* species complex [15]. This fungal group was also reported as fruit rot pathogens of red raspberries in the mid-Atlantic and co-occurrence with *Drosophila suzukii* [52], which provided a clue to clarify the association between our strains and diseased plant samples.

*Cladosporium* spp. are distributed widely as saprobic or endophytic fungi, which are often isolated from air, soil, textiles or many other substrates. Occasionally, they occur as opportunistic pathogens invading the dead or rotten issues of many plants [53]. For example, *C. sphaerospermum* has been reported to cause diseases of *Aloe vera* in India, and one *Cladosporium* sp. can cause strawberry rot in Brazil [54,55]. Bautista et al. identified *C. cladosporioides* as a microorganism associated with the anthracnose of Musa paradisiaca in the Philippines [56]. All our strains were isolated from diseased samples of ten plant hosts, and four taxa (*C. congjiangedsis*, C. ribus, *C. subuliforme* and *C. tenuissimum*) were found on two plant hosts in the meantime. At the same time, different *Cladosporium* taxa can also be discovered on one plant sample, similar to the previous study on *Eucommia ulmoides* [20]. We believed *Cladosporium* spp. on *Passiflora edulis* were able to cause one leaf blight symptom but belonged to an opportunistic pathogen because it was only found in greenhouse environments with a high temperature and humidity. Because of abundant plant diversity in Guizhou Province, after comprehensive investigation, there will be an overwhelming number and diversity of *Cladosporium* spp. and other fungi.

## Figures and Tables

**Figure 1 jof-09-00250-f001:**
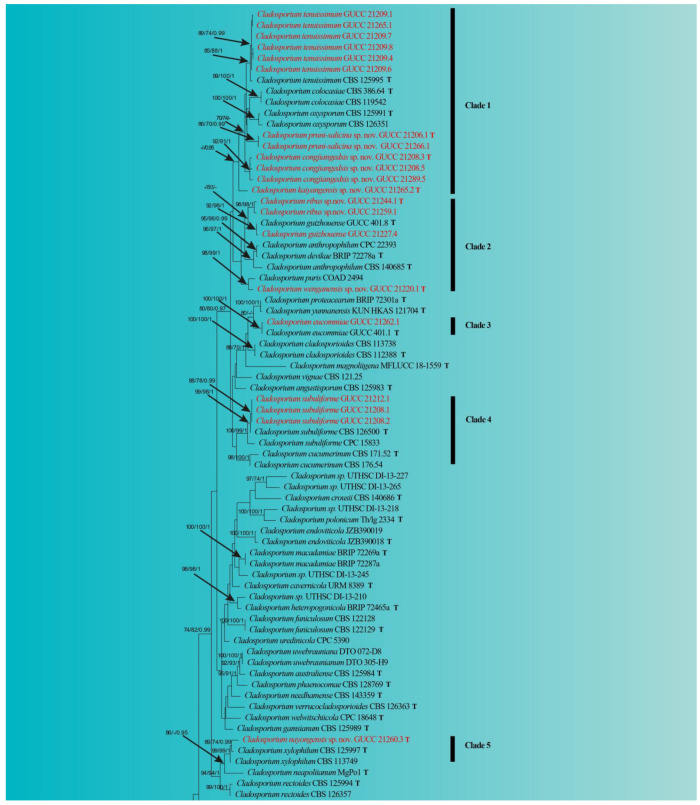
Maximum likelihood (RAxML) tree from the combined analysis of ITS, *tef*1-ɑ and *act* sequences of *Cladosporium* taxa. The tree was rooted with *Toxicocladosporium irritans* (CBS 185.58) and *T. protearum* (CBS 126499). ML and MP bootstrap values (above 70%) and Bayesian posterior probability (above 0.95) are indicated along branches (ML/MP/PP). Our isolates are highlighted in red. T = ex-type strain.

**Figure 2 jof-09-00250-f002:**
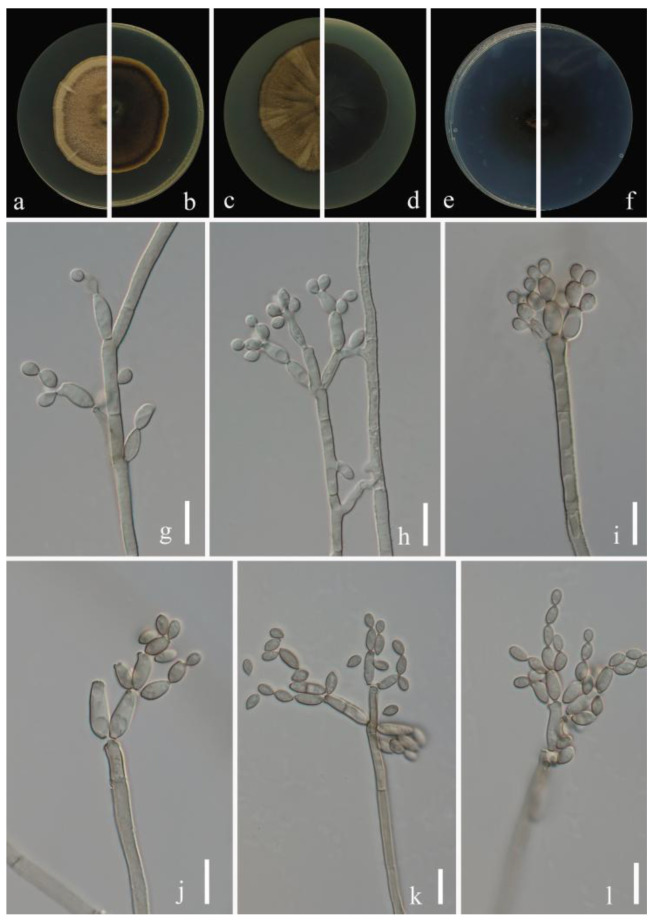
***Cladosporium pruni-salicina*** (HGUP 21206, holotype). (**a**,**b**) Culture on PDA from above and reverse. (**c**,**d**) Culture on MEA from above and reverse. (**e**,**f**) Culture on SNA from above and reverse. (**g**–**l**) conidiophores, secondary ramoconidia and conidia on SNA. Scale bars = 10 µm.

**Figure 3 jof-09-00250-f003:**
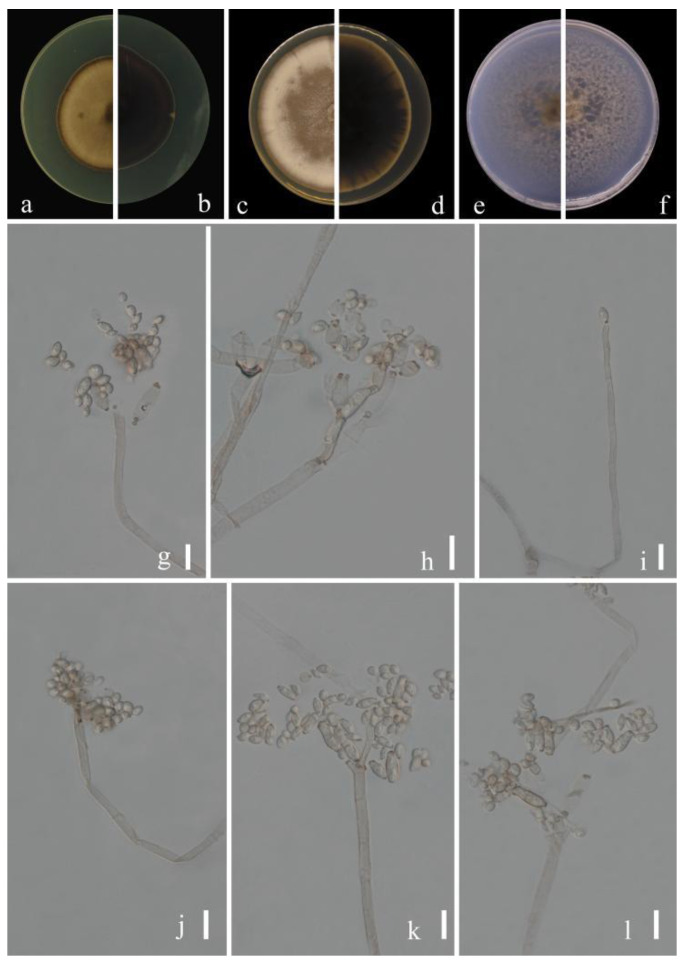
***Cladosporium congjiangedsis*** (HGUP 21208, holotype). (**a**,**b**) Culture on PDA from above and reverse. (**c**,**d**) Culture on MEA from above and reverse. (**e**,**f**) Culture on SNA from above and reverse. (**g**–**l**) Conidiophores, secondary ramoconidia and conidia on SNA. Scale bars = 10 µm.

**Figure 4 jof-09-00250-f004:**
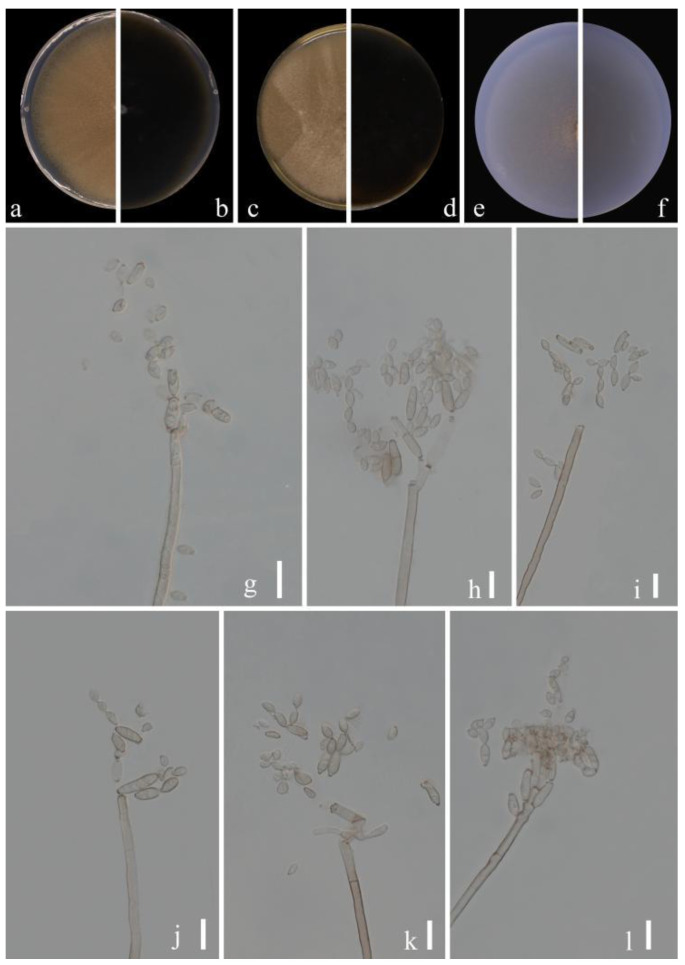
***Cladosporium kaiyangensis*** (HGUP 21265, holotype). (**a**,**b**) Culture on PDA from above and reverse. (**c**,**d**) Culture on MEA from above and reverse. (**e**,**f**) Culture on SNA from above and reverse. (**g**–**l**) Conidiophores, secondary ramoconidia and conidia on SNA. Scale bars = 10 µm.

**Figure 5 jof-09-00250-f005:**
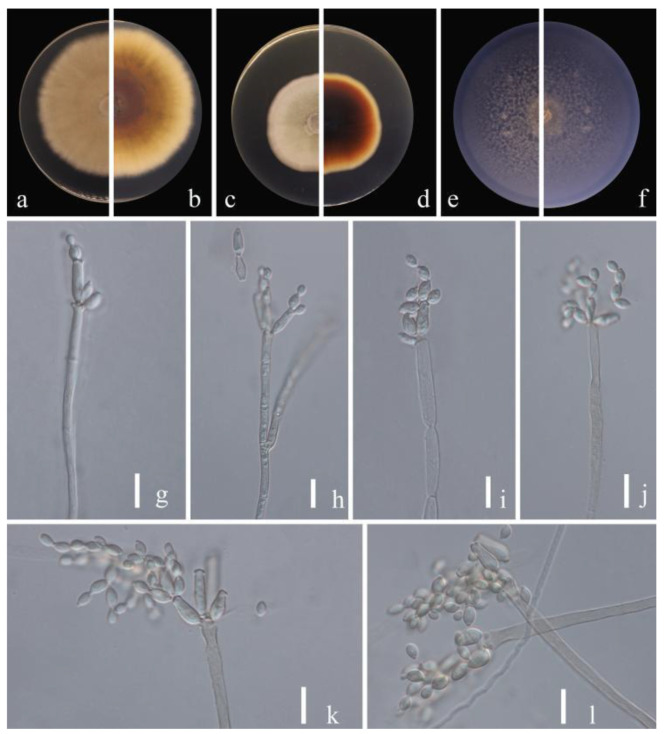
***Cladosporium ribus*** (HGUP 21244, holotype). (**a**,**b**) Culture on PDA from above and reverse. (**c**,**d**) Culture on MEA from above and reverse. (**e**,**f**) Culture on SNA from above and reverse. (**g**–**l**) Conidiophores, secondary ramoconidia and conidia on SNA. Scale bars = 10 µm.

**Figure 6 jof-09-00250-f006:**
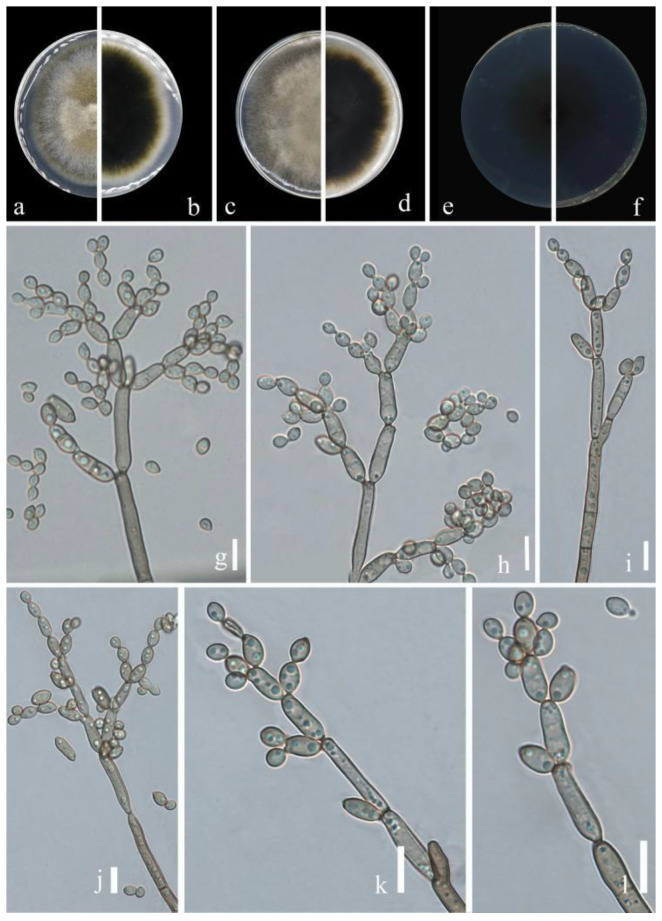
***Cladosporium wenganensis*** (HGUP 21220, holotype). (**a**,**b**) Culture on PDA from above and reverse. (**c**,**d**) Culture on MEA from above and reverse. (**e**,**f**) Culture on SNA from above and reverse. (**g**–**l**) Conidiophores, secondary ramoconidia and conidia on SNA. Scale bars = 10 µm.

**Figure 7 jof-09-00250-f007:**
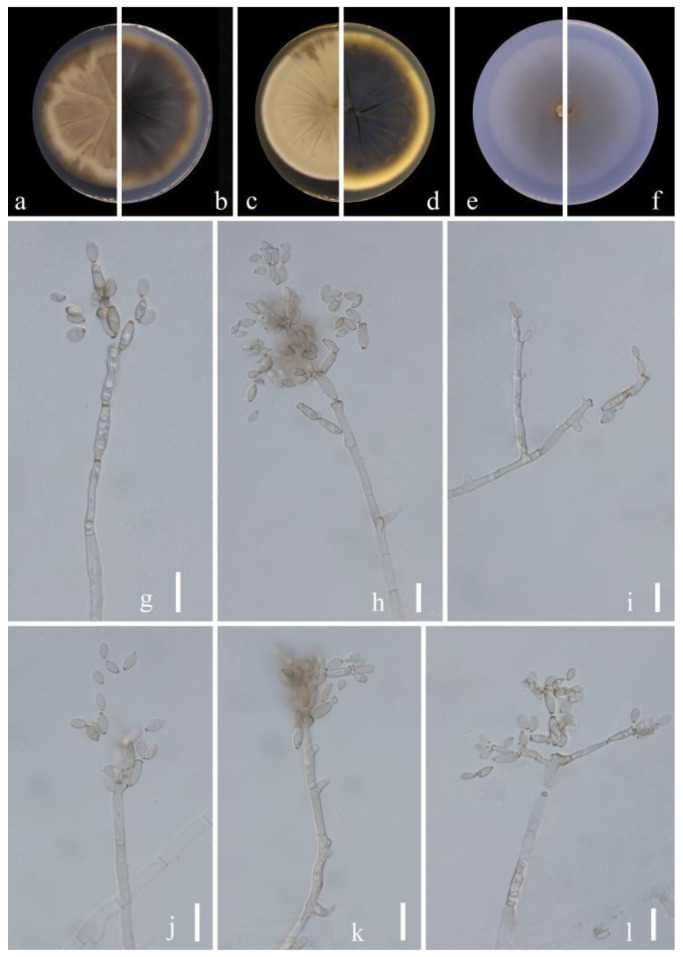
***Cladosporium nayongensis*** (HGUP 21260, holotype). (**a**,**b**) Culture on PDA from above and reverse. (**c**,**d**) Culture on MEA from above and reverse. (**e**,**f**) Culture on SNA from above and reverse. (**g**–**l**) Conidiophores, secondary ramoconidia and conidia on SNA. Scale bars = 10 µm.

**Figure 8 jof-09-00250-f008:**
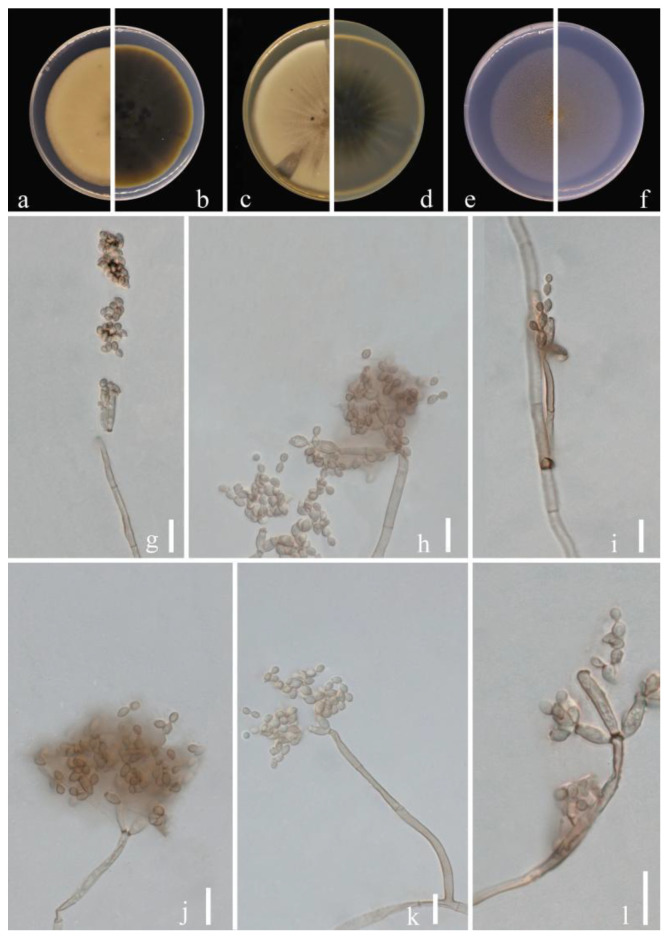
***Cladosporium punicae*** (HGUP 21271, holotype). (**a**,**b**) Culture on PDA from above and reverse. (**c**,**d**) Culture on MEA from above and reverse. (**e**,**f**) Culture on SNA from above and reverse. (**g**–**l**) Conidiophores, secondary ramoconidia and conidia on SNA. Scale bars: = 10 µm.

**Figure 9 jof-09-00250-f009:**
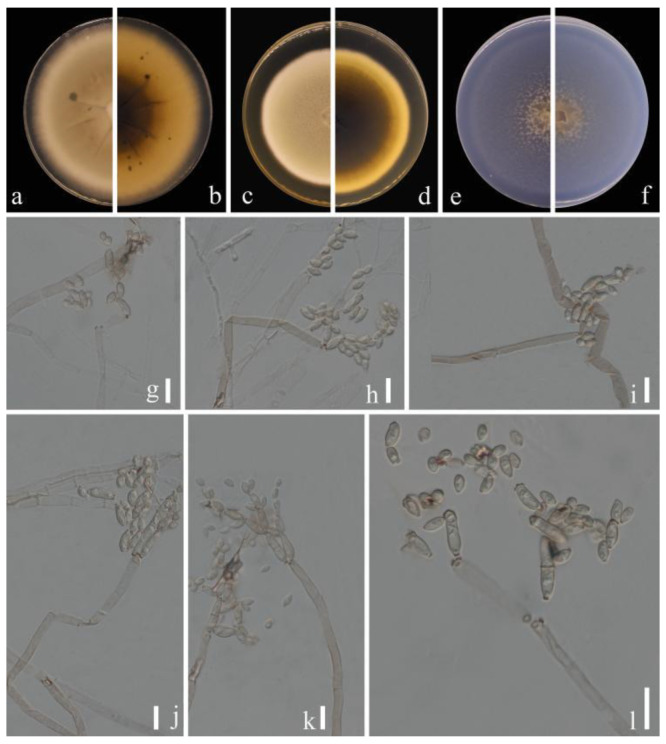
***Cladosporium subuliforme*** (GUCC 21208.1). (**a**,**b**) Culture on PDA from above and reverse. (**c**,**d**) Culture on MEA from above and reverse. (**e**,**f**) Culture on SNA from above and reverse. (**g**–**l**) Conidiophores, secondary ramoconidia and conidia on SNA. Scale bars = 10 µm.

**Figure 10 jof-09-00250-f010:**
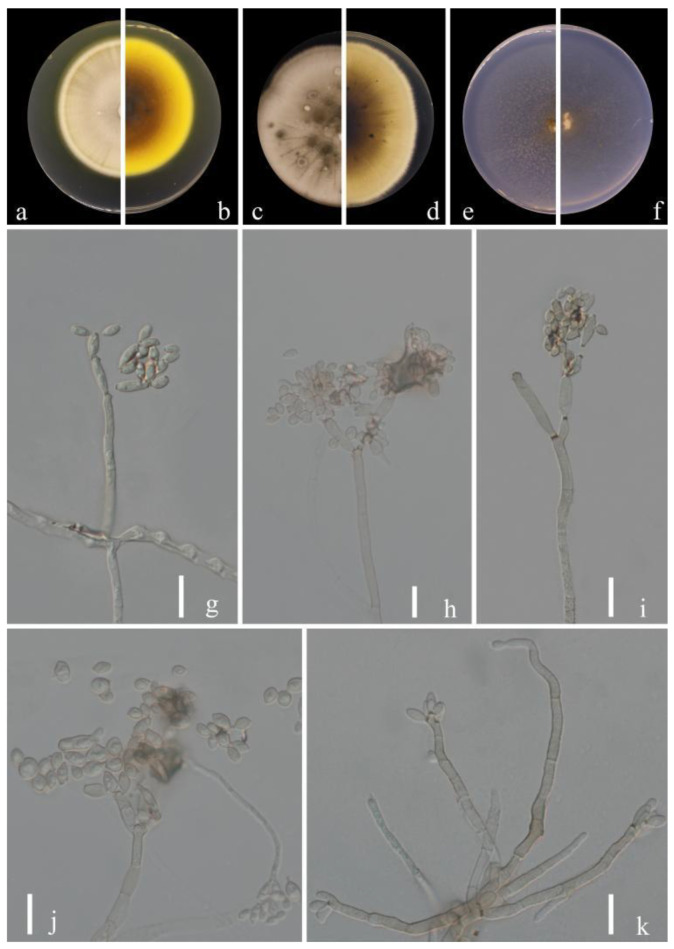
***Cladosporium xanthochromaticum*** (GUCC 21267.1). (**a**,**b**) Culture on PDA from above and reverse. (**c**,**d**) Culture on MEA from above and reverse. (**e**,**f**) Culture on SNA from above and reverse. (**g**–**k**) Conidiophores, secondary ramoconidia and conidia on SNA. Scale bars = 10 µm.

**Table 1 jof-09-00250-t001:** *Cladosporium* species studied from Guizhou Province, China with GenBank numbers. Ex-type species are in bold.

Species	Isolate	Locality	GenBank Accession Number		
			ITS	*tef*1-ɑ	*act*
** *C. congjiangedsisi* **	**GUCC 21208.3**	**Congjiang County**	**OP852675**	**OP859042**	**OP863094**
*C. congjiangedsisi*	GUCC 21208.5	Congjiang County	OP852676	OP859043	OP863095
*C. congjiangedsisi*	GUCC 21289.5	Luodian County	OP852667	OP859044	OP863096
*C. eucommiae*	GUCC 21262.1	Nayong County	OP852670	OP859050	OP863102
*C. guizhouense*	GUCC 21227.4	Kaiyang County	OP852663	OP859048	OP863100
** *C. kaiyangensis* **	**GUCC 21265.2**	**Kaiyang County**	**OP852665**	**OP859045**	**OP863097**
** *C. nayongensis* **	**GUCC 21260.3**	**Nayong County**	**OP852669**	**OP859054**	**OP863106**
** *C. pruni-salicina* **	**GUCC 21206.1**	**Kaiyang County**	**OP852683**	**OP859041**	**OP863092**
*C. pruni-salicina*	GUCC 21266.1	Kaiyang County	OP852684	-	OP863093
** *C. punicae* **	**GUCC 21271.5**	**Wengan County**	**OP852672**	**OP859056**	**OP863108**
** *C. ribus* **	**GUCC 21244.1**	**Longli County**	**OP852666**	**OP859046**	**OP863098**
*C. ribus*	GUCC 21259.1	Nayong County	OP852668	OP859047	OP863099
*C. subuliforme*	GUCC 21208.1	Congjiang County	OP852673	OP859051	OP863103
*C. subuliforme*	GUCC 21208.2	Congjiang County	OP852674	OP859052	OP863104
*C. subuliforme*	GUCC 21212.1	Dejiang County	OP852662	OP859053	-
*C. tenuissimum*	GUCC 21209.1	Congjiang County	OP852677	OP859036	OP863087
*C. tenuissimum*	GUCC 21209.4	Congjiang County	OP852678	OP859037	OP863088
*C. tenuissimum*	GUCC 21209.6	Congjiang County	OP852679	OP859038	-
*C. tenuissimum*	GUCC 21209.7	Congjiang County	OP852680	OP859039	OP863090
*C. tenuissimum*	GUCC 21209.8	Congjiang County	OP852681	OP859040	OP863091
*C. tenuissimum*	GUCC 21265.1	Kaiyang County	OP852664	OP859035	OP863086
** *C. wenganensis* **	**GUCC 21220.1**	**Wengan County**	**OP852682**	**OP859049**	**OP863101**
*C. xanthochromaticum*	GUCC 21267.1	Wengan County	OP852671	OP859055	OP863107

**Table 2 jof-09-00250-t002:** *Cladosporium* strains and their corresponding DNA sequences, which were used in the phylogenetic analyses (ex-type strains are indicated in bold).

Species Name	Strain Number	GenBank Accession Number			Reference
		ITS	*tef*1-ɑ	*act*	
** *Cladosporium acalyphae* **	**CBS 125982**	**HM147994**	**HM148235**	**HM148481**	**Bensch et al.** [15]
** *C. alboflavescens* **	**CBS 140690**	**LN834420**	**LN834516**	**LN834604**	**Sandoval-Denis et al.** [17]
** *C. angulosum* **	**CBS 140692**	**LN834425**	**LN834521**	**LN834609**	**Sandoval-Denis et al.** [17]
*C. angulosum*	COAD 2500	MK253346	MK293786	MK249989	Freitas et al. [32]
** *C. angustisporum* **	**CBS 125983**	**HM147995**	**HM148236**	**HM148482**	**Bensch et al.** [15]
** *C. angustiterminale* **	**CBS 140480**	**KT600379**	**KT600476**	**KT600575**	**Bensch et al.** [16]
** *C. anthropophilum* **	**CBS 140685**	**LN834437**	**LN834533**	**LN834621**	**Sandoval-Denis et al.** [33]
*C. anthropophilum*	CPC 22393	MF472922	MF473349	MF473772	Bensch et al. [7]
** *C. aphidis* **	**CPC 13204**	**JN906978**	**JN906984**	**JN906997**	**Bensch et al.** [6]
** *C. arenosum* **	**CHFC-EA 566**	**MN879328**	**MN890011**	**MN890008**	**Crous et al.** [34]
** *C. asperulatum* **	**CBS 126340**	**HM147998**	**HM148239**	**HM148485**	**Bensch et al.** [15]
** *C. australiense* **	**CBS 125984**	**NR_119837**	**HM148240**	**HM148486**	**Bensch et al.** [15]
** *C. austroafricanum* **	**CBS 140481**	**KT600381**	**KT600478**	**KT600577**	**Bensch et al.** [16]
** *C. austrolitorale* **	**CBS 148321**	**MN879327**	**MN890010**	**MN890007**	**Crous et al.** [34]
** *C. caprifimosum* **	**FMR 16532**	**LR813198**	**LR813210**	**LR813205**	**Isabel Iturrieta-González et al.** [19]
** *C. cavernicola* **	**URM 8389**	**MZ518829**	**MZ555733**	**MZ555746**	**Pereira et al.** [35]
** *C. chalastosporoides* **	**CBS 125985**	**HM148001**	**HM148242**	**HM148488**	**Bensch et al.** [15]
** *C. chasmanthicola* **	**CPC 21300**	**NR_152307**	**KY646227**	**KY646224**	**Marin-Felix et al.** [18]
** *C. chubutense* **	**CBS 124457**	**FJ936158**	**FJ936161**	**FJ936165**	**Schubert et al.** [36]
** *C. cladosporioides* **	**CBS 112388**	**NR_119839**	**HM148244**	**HM148490**	**Bensch et al.** [15]
*C. cladosporioides*	CBS 113738	HM148004	HM148245	HM148491	Bensch et al. [15]
** *C. colocasiae* **	**CBS 386.64**	**HM148067**	**HM148310**	**HM148555**	**Bensch et al.** [15]
*C. colocasiae*	CBS 119542	HM148066	HM148309	HM148554	Bensch et al. [15]
** *C. colombiae* **	**CBS 274.80B**	**FJ936159**	**FJ936163**	**FJ936166**	**Schubert et al.** [36]
** *C. coprophilum* **	**FMR 16164**	**LR813201**	**LR813213**	**LR813207**	**Isabel Iturrieta-González et al.** [19]
** *C. crousii* **	**CBS 140686**	**LN834431**	**LN834527**	**LN834615**	**Sandoval-Denis et al.** [33]
** *C. cucumerinum* **	**CBS 171.52**	**NR_119841**	**HM148316**	**HM148561**	**Bensch et al.** [15]
*C. cucumerinum*	CBS 176.54	HM148078	HM148322	HM148567	Bensch et al. [15]
** *C. delicatulum* **	**CBS 126344**	**HM148081**	**HM148325**	**HM148570**	**Bensch et al.** [15]
** *C. devikae* **	**BRIP 72278a**	**MZ303808**	**MZ344193**	**MZ344212**	**Prasannath et al.** [37]
** *C. endoviticola* **	**JZB390018**	**-**	**MN984228**	**MN984220**	**Manawasinghe et al.** [38]
*C. endoviticola*	JZB390019	-	MN984229	MN984221	Manawasinghe et al. [38]
** *C. eucommiae* **	**GUCC 401.1**	**OL587465**	**OL504966**	**OL519775**	**Wang et al.** [20]
** *C. europaeum* **	**CBS 134914**	**HM148056**	**HM148298**	**HM148543**	**Bensch et al.** [15]
*C. europaeum*	FP-027-A9	MH102078	MH102121	MH102068	Patyshakuliyeva et al. [39]
** *C. exasperatum* **	**CBS 125986**	**HM148090**	**HM148334**	**HM148579**	**Bensch et al.** [15]
** *C. exile* **	**CBS 125987**	**HM148091**	**HM148335**	**HM148580**	**Bensch et al.** [15]
** *C. flabelliforme* **	**CBS 126345**	**HM148092**	**HM148336**	**HM148581**	**Bensch et al.** [15]
** *C. flavovirens* **	**CBS 140462**	**LN834440**	**LN834536**	**LN834624**	**Sandoval-Denis et al.** [17]
** *C. funiculosum* **	**CBS 122129**	**NR_119845**	**HM148338**	**HM148583**	**Bensch et al.** [15]
*C. funiculosum*	CBS 122128	HM148093	HM148337	HM148582	Bensch et al. [15]
*C. fuscoviride*	FMR 16385	LR813200	LR813212	LR813206	Isabel Iturrieta-González et al. [19]
** *C. gamsianum* **	**CBS 125989**	**HM148095**	**HM148339**	**HM148584**	**Bensch et al.** [15]
** *C. globisporum* **	**CBS 812.96**	**HM148096**	**HM148340**	**HM148585**	**Bensch et al.** [15]
** *C. grevilleae* **	**CBS 114271**	**JF770450**	**JF770472**	**JF770473**	**Crous et al.** [40]
** *C. guizhouense* **	**GUCC 401.8**	**ON334728**	**ON383470**	**ON383338**	**Wang et al.** [20]
** *C. heteropogonicola* **	**BRIP 72465a**	**OL307932**	**OL332742**	**OL332743**	**Tan et al.** [41]
** *C. hillianum* **	**CBS 125988**	**HM148097**	**HM148341**	**HM148586**	**Bensch et al.** [15]
** *C. inversicolor* **	**CBS 401.80**	**HM148101**	**HM148345**	**HM148590**	**Bensch et al.** [15]
** *C. ipereniae* **	**CBS 140483**	**KT600394**	**KT600491**	**KT600589**	**Bensch et al.** [16]
** *C. iranicum* **	**CBS 126346**	**HM148110**	**HM148354**	**HM148599**	**Bensch et al.** [15]
*C. kenpeggii*	CPC 19248	KY646222	KY646228	KY646225	**Marin-Felix et al. [18]**
** *C. lentulum* **	**FMR 16288**	**LR813203**	**LR813215**	**LR813209**	**Isabel Iturrieta-González et al.** [19]
** *C. licheniphilum* **	**CBS 125990**	**HM148111**	**HM148355**	**HM148600**	**Bensch et al.** [15]
** *C. longicatenatum* **	**CBS 140485**	**KT600403**	**KT600500**	**KT600598**	**Bensch et al.** [16]
*C. lycoperdinum*	CBS 126347	HM148112	HM148356	HM148601	Bensch et al. [15]
*C. lycoperdinum*	CBS 574.78C	HM148115	HM148359	HM148604	Bensch et al. [15]
** *C. macadamiae* **	**BRIP 72269a**	**MZ303810**	**MZ344195**	**MZ344214**	**Prasannath et al.** [37]
*C. macadamiae*	BRIP 72287a	MZ303811	MZ344196	MZ344215	Prasannath et al. [37]
** *C. magnoliigena* **	**MFLUCC 18-1559**	**MK347813**	**MK340864**	**-**	**Jayasiri et al.** [42]
*C. montecillanum*	CPC 15605	KT600407	KT600505	KT600603	Bensch et al. [16]
** *C. montecillanum* **	**CBS 140486**	**KT600406**	**KT600504**	**KT600602**	**Bensch et al.** [16]
** *C. myrtacearum* **	**CBS 126350**	**HM148117**	**HM148361**	**HM148606**	**Bensch et al.** [15]
*C. myrtacearum*	CBS 126349	MH863925	HM148360	HM148605	Vu et al. [43]
** *C. neapolitanum* **	**MgPo1**	**MK387890**	**MK416094**	**MK416051**	**Zimowska et al.** [44]
** *C. needhamense* **	**CBS 143359**	**MF473142**	**MF473570**	**MF473991**	**Bensch et al.** [7]
** *C. neopsychrotolerans* **	**CGMCC 3.18031**	**KX938383**	**KX938400**	**KX938366**	**Ma et al.** [45]
** *C. oxysporum* **	**CBS 125991**	**NR_152267**	**HM148362**	**HM148607**	**Bensch et al.** [15]
*C. oxysporum*	CBS 126351	MH863927	HM148363	HM148608	Vu et al. [43]
** *C. paracladosporioides* **	**CBS 171.54**	**HM148120**	**HM148364**	**HM148609**	**Bensch et al.** [15]
** *C. parapenidielloides* **	**CBS 140487**	**KT600410**	**KT600508**	**KT600606**	**Bensch et al.** [16]
** *C. perangustum* **	**CBS 125996**	**HM148121**	**HM148365**	**HM148610**	**Bensch et al.** [15]
** *C. phaenocomae* **	**CBS 128769**	**JF499837**	**JF499875**	**JF499881**	**Crous et al.** [46]
** *C. phyllactiniicola* **	**CBS 126355**	**NR_111537**	**HM148397**	**HM148642**	**Bensch et al.** [15]
** *C. phyllophilum* **	**CBS 125992**	**HM148154**	**HM148398**	**HM148643**	**Bensch et al.** [15]
** *C. pini-ponderosae* **	**CBS 124456**	**FJ936160**	**FJ936164**	**FJ936167**	**Schubert et al.** [36]
** *C. polonicum* **	**Th/lg/2334**	**MK387894**	**MK416098**	**MK416055**	**Zimowska et al.** [44]
** *C. proteacearum* **	**BRIP 72301a**	**MZ303809**	**MZ344194**	**MZ344213**	**Prasannath et al.** [37]
** *C. pseudochalastosporoides* **	**CBS 140490**	**NR_152296**	**KT600513**	**KT600611**	**Bensch et al.** [16]
*C. puris*	COAD 2494	MK253338	MK293778	MK249981	Freitas et al. [32]
** *C. queenslandicum* **	**BRIP 72447a**	**OL307928**	**OL332735**	**OL332736**	**Tan et al.** [41]
** *C. rectoides* **	**CBS 125994**	**HM148193**	**HM148438**	**HM148683**	**Bensch et al.** [15]
*C. rectoides*	CBS 126357	MH863933	HM148439	HM148684	Vu et al. [43]
*C. rubrum*	CMG 28	MN053018	MN066644	MN066639	Vicente et al. [47]
** *C. ruguloflflabelliforme* **	**CBS 140494**	**KT600458**	**KT600557**	**KT600655**	**Bensch et al.** [16]
** *C. rugulovarians* **	**CBS 140495**	**KT600459**	**KT600558**	**KT600656**	**Bensch et al.** [16]
** *C. scabrellum* **	**CBS 126358**	**HM148195**	**HM148440**	**HM148685**	**Bensch et al.** [15]
** *C. silenes* **	**CBS 109082**	**EF679354**	**EF679429**	**EF679506**	**Schubert et al.** [14]
** *C. sinuatum* **	**CGMCC 3.18096**	**KX938385**	**KX938402**	**KX938368**	**Ma et al.** [45]
*Cladosporium* sp.	UTHSC DI-13-227	LN834422	LN834518	LN834606	Sandoval-Denis et al. [33]
*Cladosporium* sp.	UTHSC DI-13-245	LN834429	LN834525	LN834613	Sandoval-Denis et al. [33]
*Cladosporium* sp.	UTHSC DI-13-265	LN834435	LN834531	LN834619	Sandoval-Denis et al. [33]
*Cladosporium* sp.	UTHSC DI-13-218	LN834418	LN834514	LN834602	Sandoval-Denis et al. [33]
*Cladosporium* sp.	UTHSC DI-13-210	LN834414	LN834510	LN834598	Sandoval-Denis et al. [33]
** *C. stipagrostidicola* **	**CBS 146978**	**MZ064420**	**MZ078223**	**MZ078146**	**Crous et al.** [48]
** *C. subuliforme* **	**CBS 126500**	**NR_119854**	**HM148441**	**HM148686**	**Bensch et al.** [15]
*C. subuliforme*	CPC 15833	KT600453	KT600552	KT600650	Bensch et al. [16]
** *C. tenuissimum* **	**CBS 125995**	**HM148197**	**HM148442**	**HM148687**	**Bensch et al.** [15]
** *C. tianshanense* **	**CGMCC 3.18033**	**KX938381**	**KX938398**	**KX938364**	**Ma et al.** [45]
*C. uredinicola*	CPC 5390	AY251071	HM148467	HM148712	Braun et al. [13]
*C. uwebrauniana*	DTO 072-D8	MF473306	MF473729	MF474156	Bensch et al. [7]
*C. uwebraunianum*	DTO 305-H9	MF473307	MF473730	MF474157	Bensch et al. [7]
** *C. varians* **	**CBS 126362**	**HM148224**	**HM148470**	**HM148715**	**Bensch et al.** [15]
** *C. verrucocladosporioides* **	**CBS 126363**	**HM148226**	**HM148472**	**HM148717**	**Bensch et al.** [15]
*C. vicinum*	CPC 22316	MF473311	MF473734	MF474161	Bensch et al. [7]
*C. vignae*	CBS 121.25	HM148227	HM148473	HM148718	Bensch et al. [15]
** *C. welwitschiicola* **	**CPC 18648**	**NR_152308**	**KY646229**	**KY646226**	**Marin-Felix et al.** [18]
** *C. westerdijkiae* **	**CBS 113746**	**HM148061**	**HM148303**	**HM148548**	**Bensch et al.** [15]
** *C. xanthochromaticum* **	**CBS 140691**	**LN834415**	**LN834511**	**LN834599**	**Sandoval-Denis et al.** [33]
*C. xanthochromaticum*	CBS 126364	HM148122	HM148366	HM148611	Bensch et al. [15]
** *C. xylophilum* **	**CBS 125997**	**NR_111541**	**HM148476**	**HM148721**	**Bensch et al.** [15]
*C. xylophilum*	CBS 113749	HM148228	HM148474	HM148719	Bensch et al. [15]
** *C. yunnanensis* **	**KUN HKAS 121704**	**OK338502**	**OL825680**	**OL466937**	**Xu et al.** [49]
** *Toxicocladosporium irritans* **	**CBS 185.58**	**NR_152316**	**-**	**LT821375**	**Crous et al.** [50]
** *T. protearum* **	**CBS 126499**	**NR_152321**	**-**	**LT821379**	**Crous et al.** [51]

**Table 3 jof-09-00250-t003:** DNA base differences between our strains and related taxa in the three gene regions (including gaps). Asterisks (*) denote our material, (T) = ex-type strain.

Species	Strain Number	Gene Region and Alignment Positions		
		*act* (1–229 Characters)	ITS (230–769 Characters)	*tef*1-ɑ (770–1018 Characters)
*Cladosporium tenuissimum*	CBS 125995^T^	-	-	-
*C. tenuissimum* *	GUCC 21265.1	4	2	5
*C. tenuissimum* *	GUCC 21209.1	4	2	5
*C. tenuissimum* *	GUCC 21209.4	4	1	5
*C. tenuissimum* *	GUCC 21209.6	N/A	0	5
*C. tenuissimum* *	GUCC 21209.7	3	0	5
*C. tenuissimum* *	GUCC 21209.8	4	0	5
		***act* (1–232 characters)**	**ITS (233–768 characters)**	** *tef* ** **1-** **ɑ (769–1087 characters)**
*C. colocasiae*	CBS 386.64^T^	-	-	-
*C. colocasiae*	CBS 119542	0	0	1
*C. oxysporum*	CBS 125991^T^	17	0	13
*C. oxysporum*	CBS 126351	17	1	21
*C. pruni-salicina* sp. nov. *	GUCC 21206.1^T^	12	1	30
*C. pruni-salicina* sp. nov. *	GUCC 21266.1	12	0	N/A
*C. congjiangedsis* sp. nov. *	GUCC 21208.3^T^	14	0	22
*C. congjiangedsis* sp. nov. *	GUCC 21208.5	13	0	22
*C. congjiangedsis* sp. nov. *	GUCC 21289.5	13	0	22
*C. kaiyangensis* sp. nov. *	GUCC 21265.2^T^	14	3	29
		***act* (1–226 characters)**	**ITS (227–765 characters)**	** *tef* ** **1-** **ɑ (766–1061 characters)**
*C. oxysporum*	CBS 125991^T^	-	-	-
*C. oxysporum*	CBS 126351	0	1	5
*C. pruni-salicina* sp. nov. *	GUCC 21206.1^T^	16	1	21
*C. pruni-salicina* sp. nov. *	GUCC 21266.1	16	0	N/A
		***act* (1–232 characters)**	**ITS (233–771 characters)**	** *tef* ** **1-** **ɑ (772–1007 characters)**
*C. congjiangedsis* sp. nov. *	GUCC 21208.3^T^	-	-	-
*C. congjiangedsis* sp. nov. *	GUCC 21208.5	3	0	3
*C. congjiangedsis* sp. nov. *	GUCC 21289.5	7	0	0
*C. tenuissimum*	CBS 125995^T^	8	0	17
*C. oxysporum*	CBS 125991^T^	18	0	16
*C. oxysporum*	CBS 126351	18	1	23
*C. pruni-salicina* sp. nov. *	GUCC 21206.1^T^	4	1	28
*C. pruni-salicina* sp. nov. *	GUCC 21266.1	4	0	N/A
		***act* (1–232 characters)**	**ITS (233–773 characters)**	** *tef* ** **1-** **ɑ (774–1023 characters)**
*C. kaiyangensis*	GUCC 21265.2^T^	-	-	-
*C. tenuissimum*	CBS 125995^T^	3	4	25
*C. colocasiae*	CBS 386.64^T^	16	7	28
*C. oxysporum*	CBS 125991^T^	18	4	21
*C. pruni-salicina* sp. nov. *	GUCC 21206.1^T^	4	5	35
*C. congjiangedsis* sp. nov. *	GUCC 21208.3^T^	4	4	19
		***act* (1–232 characters)**	**ITS (233–770 characters)**	** *tef* ** **1-** **ɑ (771–1016 characters)**
*C. guizhouense*	GUCC 401.8^T^	-	-	-
*C. guizhouense* *	GUCC 21227.4	0	0	4
*C. ribus* sp. nov. *	GUCC 21244.1^T^	4	0	20
*C. ribus* sp. nov. *	GUCC 21259.1	4	0	20
		***act* (1–183 characters)**	**ITS (184–685 characters)**	** *tef* ** **1-** **ɑ (686–887 characters)**
*C. puris*	COAD 2494	-	-	-
*C. wenganensis* sp. nov. *	GUCC 21220.1^T^	7	1	10
		***act* (1–232 characters)**	**ITS (233–773 characters)**	** *tef* ** **1-** **ɑ (774–1021 characters)**
*C. eucommiae*	GUCC 401.1^T^	-	-	-
*C. eucommiae* *	GUCC 21262.1	0	1	2
		***act* (1–226 characters)**	**ITS (227–743 characters)**	** *tef* ** **1-** **ɑ (744–1063 characters)**
*C. subuliforme*	CBS 126500^T^	-	-	-
*C. subuliforme*	CPC 15833	9	0	5
*C. subuliforme* *	GUCC 21212.1	N/A	0	1
*C. subuliforme* *	GUCC 21208.1	0	0	1
*C. subuliforme* *	GUCC 21208.2	0	0	1
		***act* (1–226 characters)**	**ITS (267–743 characters)**	** *tef* ** **1-** **ɑ (744–1062 characters)**
*C. xylophilum*	CBS 125997	-	-	-
*C. xylophilum*	CBS 113749	4	0	1
*C. nayongensis* sp. nov. *	GUCC 21260.3^T^	10	0	2
		***act* (1–266 characters)**	**ITS (267–732 characters)**	** *tef* ** **1-** **ɑ (733–973 characters)**
*C. xanthochromaticum*	CBS 140691^T^	-	-	-
*C. xanthochromaticum*	CBS 126364	5	3	26
*C. xanthochromaticum* *	GUCC 21267.1	0	3	3
		***act* (1–229 characters)**	**ITS (230–770 characters)**	** *tef* ** **1-** **ɑ (771–1018 characters)**
*C. perangustum*	CBS 125996^T^	-	-	-
*C. punicae* sp. nov. *	GUCC 21271.5^T^	7	2	31

## Data Availability

All data generated or analyzed during this study are included in this published article and/or are available from the corresponding author upon reasonable request.

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
