# Peer review of "Cladosporium Species Associated with Fruit Trees in Guizhou Province, China"

_jof, 2023, doi:10.3390/jof9020250_

Round 1
Reviewer 1 Report
Manuscript "Cladosporium Species Associated with Fruit Trees in Guizhou Province, China" present the fungal diversity of Cladosporium species on named host trees in China.
Cladosporium strains (23) were isolated, and described in cultures according to morphology. Isolates were characterized by 3 relevant genetic markers. and molecular phylogenetic analysis were done.
7 new Cladosporium species were described. All is documented and presented with well and nice photos, and solid descriptions. And it is well supported with phylogenetic analysis.
This research increased knowledge about the diversity and distribution of the species of Chladosporium. It is well presented and clearly written. I found a few typo (marked in manuscript).

Author Response
Response to Reviewer 1 Comments
Point 1: In the materials and methods section, delete "2020" in line 47
Response 1: Thank you for your suggestion. In the revised copy, we have deleted "2020" in line 47.
Point 2: In the morphology section, add "space between" at line 547.
Response 2: Thank you again for your suggestion. In the revised copy, a space has been added in line 547.

Reviewer 2 Report
This manuscript identifies Cladosporium species based on morphological and phylogenetic data.
It is accurate and well written. It is suitable for publication after minor revision.
1. Materials and methods:
Line 45: It will be interesting to the readers if authors can provide fruit tree species (e.g. apple trees (Malus domestica....)
2. Strains origin and sources should be inserted in Table 1 or cite the papers.
3. Discussion: Discussion should focus more on the substrate of each species was found.
Author Response
Response to Reviewer 2 Comments
Point 1: In Materials and methods: Line 45: It will be interesting to the readers if authors can provide fruit tree species (e.g. apple trees (Malus domestica....)
Response 1: Thank for your suggestion. In the revised copy, we have added the sentence to refer to the fruit tree species "including 10 host specifications (Passionfruit, Cherry, Plum, Shaddock, Loquat, Rosa roxburghii, Walnut, Pomegranate, Tangerines and Pitaya)" in the material and method section at line 45.
Point 2: Strains origin and sources should be inserted in Table 1 or cite the papers.
Response 2: Thank again for your suggestion. In the revised copy, the source of all strains has been added in Table 1, which also cited the reference.
Point 3: Discussion: Discussion should focus more on the substrate of each species was found.
Response 3: Thank again for your suggestion. In the revised copy, we added “Cladosporium spp. are distributed widely as saprobic or endophytic fungi, which are often isolated from air, soil, textiles or many other substrates. Occasionally, they occur as the opportunistic pathogens invading on dead or rotten issues of many plants [54]. For example, C. sphaerospermum was reported to cause diseases of Aloe vera in India; and one Cladosporium sp. can cause strawberry rot in Brazil [55-66]. Bautista et al. identified C. cladosporioides as the microorganism associated with anthracnose of Musa paradisiaca. in the Philippines[57]. All our strains were isolated from diseased samples of ten plant hosts, and four taxa (C. tenuissimum, C. congjiangedsis, C. ribus and C. subuliforme) were found on two plant hosts in the meantime. At the same time, different Cladosporium taxa also can be discovered on one plant sample, like the previous study on Eucommia ulmoides[20]. We believed Cladosporium spp. on Passiflora edulis were able to cause one leaf blight symptom, but belonged to an opportunistic pathogen because only found in greenhouse environment with high temperature and humidity. Because of abundant plant diversity in Guizhou Province, after comprehensive investigation, there will be an overwhelming number and diversity of Cladosporium spp. and other fungi.” in the discussion.
